# An Octagonal Ring-shaped Parasitic Resonator Based Compact Ultrawideband Antenna for Microwave Imaging Applications

**DOI:** 10.3390/s20051354

**Published:** 2020-03-01

**Authors:** Amran Hossain, Mohammad Tariqul Islam, Ali F. Almutairi, Mandeep Singh Jit Singh, Kamarulzaman Mat, Md. Samsuzzaman

**Affiliations:** 1Faculty of Engineering and Built Environment, Universiti Kebangsaan Malaysia, Bangi 43600, Selangor, Malaysia; mandeep@ukm.edu.my (M.S.J.S.); kamarulzaman@ukm.edu.my (K.M.); samsuzzaman@ukm.edu.my (M.S.); 2Electrical Engineering Department, Kuwait University, Kuwait City 13060, Kuwait

**Keywords:** ultrawideband, microwave imaging, a microstrip patch antenna, ring-shaped resonator, parasitic elements

## Abstract

An Ultrawideband (UWB) octagonal ring-shaped parasitic resonator-based patch antenna for microwave imaging applications is presented in this study, which is constructed with a diamond-shaped radiating patch, three octagonal, rectangular slotted ring-shaped parasitic resonator elements, and partial slotting ground plane. The main goals of uses of parasitic ring-shaped elements are improving antenna performance. In the prototype, various kinds of slots on the ground plane were investigated, and especially rectangular slots and irregular zigzag slots are applied to enhance bandwidth, gain, efficiency, and radiation directivity. The optimized size of the antenna is 29 × 24 × 1.5 mm^3^ by using the FR-4 substrate. The overall results illustrate that the antenna has a bandwidth of 8.7 GHz (2.80–11.50 GHz) for the reflection coefficient S_11_ < −10 dB with directional radiation pattern. The maximum gain of the proposed prototype is more than 5.7 dBi, and the average efficiency over the radiating bandwidth is 75%. Different design modifications are performed to attain the most favorable outcome of the proposed antenna. However, the prototype of the proposed antenna is designed and simulated in the 3D simulator CST Microwave Studio 2018 and then effectively fabricated and measured. The investigation throughout the study of the numerical as well as experimental data explicit that the proposed antenna is appropriate for the Ultrawideband-based microwave-imaging fields.

## 1. Introduction

Microwave Imaging (MWI) is an identifying or discovering technique to estimate hidden objects in a medium using EM (electromagnetic) signals in microwave frequency ranges. Especially in medical imaging techniques, mammography, Magnetic Resonance Imaging (MRI), X-ray imaging, Computed Tomography (CT) and Ultrasound are standard techniques to detect cancer, tumor and malignant object in the human body. However, mammography and X-ray imaging have some limitations [1,2] such as destructive radiation, comparatively high false-negative rates, low sensitivity, increasing cancer risk due to short dose ionizing radiation [3]. Moreover, MRI is another standard method that helps physicians to assess different parts of the human body and identify the existence of particular diseases [4], but it is too costly and less effective [5]. Furthermore, the ultrasound technique is good for a certain domain, but it flops in the presence of bone and air, as well as image perfection depends on technicians [6]. Therefore, it is indispensable to develop a novel imaging method to detect cancer, tumor, etc. in the human body without harmful. Last few decades, the alternative technique that is MWI has been recommended as safe to prevailing medical imaging techniques together with mammography, X-ray, ultrasound, and MRI [7]. MWI is an innovative technique, which fascinates enormous interest in medical diagnostic areas, for instance, breast tumor detection, brain tumor detection, early-stage heart failure recognition, health observing, etc. due to its low cost, low profile, portability, and non-ionizing effects. In this methodology, antenna plays a major role as well as acts as a transceiver, in which the transmitting antenna propagates the microwaves and then microwaves travel through the human body. After that, data are composed of the receiving antenna. When microwave signals scattered from dissimilar tissue of the human body, it is possible to distinguish by MWI antenna sensors. In this domain, the radiated and scattered energy is received by the antenna sensor(s) for further processing.

The prime working procedure of MWI is to analyze the variance among the electrical characteristics of healthy tissue as well as malignant cells (e.g., breast tumor, brain tumor, etc.) of the human body. In a human body, the fluid of each organic tissue differs, which reasons diverse electrical characteristics. Additionally, the existence of ions, as well as free radicals in the malignant tissues, increases the dielectric loss gradually. Consequently, the tumor cells or cancer cells, with larger dielectric value than ordinary tissue, may be identified by examining the back-scattered signals from collected images [8]. However, MWI provides massive data rates to renovate quality full images, but it is more challenging to develop an appropriate system to produce high resolution and precise image of the scattered signals. An Ultra-wideband antenna is a suitable antenna in MWI applications due to its good characteristics [9], such as (i) eco-friendly (ii) biological friendliness (iii) remote processes and (iv) proficient to working in among high and low frequencies. In modern times, for these unique features, the researchers are motivated to design UWB antennas for MWI applications. It is remarkable that the antennas should be compact in dimension with superior gain, wider bandwidth as well as higher efficiency for an effective MWI system with better image quality and dynamic range. Until now, for MWI applications, a number of different categories of UWB antennas have been reported proposed in different aspects such as [9,10] (i) narrow bands versus wide bands, (ii) omnidirectional versus directional radiation patterns, (iii) low frequency versus high frequency, (iv) lower energy consumption versus capability to penetrate objects, (v) less significant electromagnetic radiation, and (vi) upper precision range.

With respect to the mentioned aspects, researchers have proposed a number of Ultrawideband antennas; for example, parasitic resonator-based antennas [9,11,12,13], differential Ultrawideband antennas [13], slotted UWB antennas [14], UWB elliptical antennas [15], sensor-based UWB [16], different types of directional UWB antennas [17,18,19,20], uniplanar metamaterial-based UWB antennas [21,22], various categories of Vivaldi antennas [23,24,25,26,27,28,29], several types of UWB monopole antennas [30,31,32], CPW-fed UWB antennas [33], tapered slot UWB [34] antennas, and numerous others. In the previous decade, UWB antennas have inaugurated to be castoff for MWI applications. In the current MWI applications era, it is necessary to design compact antennas with high performance, high bandwidth, and high gain. Furthermore, UWB communication is an attractive communication scheme due to its extraordinary characteristics (i.e., data transmission rate is high, less interference, and less spectral power density) in wireless communication [9]. Several researchers have proposed different techniques to enhance the UWB antenna’s performance such as improved pattern of patch [29], use of a metamaterials coating [35], decamped ground plane [36], adding extra different shape slots [14,25,32], and both in radiating patch and ground plane. Until now, due to various factors, a compact UWB antenna design is still a challenge. However, a parasitic resonator-based UWB antenna is presented in [9], operating from 2.7 GHz to10.3 GHz with a compact size of 30 × 25 × 1.6 mm^3^. However, in this approach, at the lower frequency, the overall gain is comparatively low, while at a high frequency the gain was up to 5.5 dBi and the antenna produced only one resonance frequency under −10 dB. A slot resonator Y-shaped UWB antenna is demonstrated in [11], and its operating frequency is 2.86 GHz to 13.3 GHz with a size of 36 × 38 × 1.6 mm^3^. It achieves higher bandwidth, but its gain is comparatively very low at a high frequency as well as it lacks suitability for high-resolution imaging. A rectangular resonator-based antenna with an operating frequency of 3.04 GHz to 11.43 GHz with a compact dimension of 14 × 22 × 1.5 mm^3^ is proposed in [14], which is a comparatively small size antenna, but the gain is not satisfactory. The authors in [26] proposed a UWB Vivaldi antenna which operates from 3.01 GHz to 11 GHz. It has a high gain at a higher frequency, but is a large size antenna. The authors in [27] proposed a UWB antipodal Vivaldi antenna, with a size of 40 × 40 × 1.6 mm^3^ and an operating frequency of 2.5 GHz to 11 GHz. This antenna is suitable for microwave imaging and its gain was comparatively higher than other Vivaldi antennas but produced an image quality comparatively less in terms of resolution.

In this paper, we propose a new compact octagonal ring-shaped parasitic resonator-based UWB antenna for microwave imaging areas. The innovations of the proposed antenna are: (i) Higher gain, wider bandwidth, directive radiation pattern, as well as lower group delay concerning recently proposed parasitic resonator-based antennas. In general, these features are prerequisites for MWI applications. (ii) It is applicable for the multiband (i.e., Wideband, C-band, X-band) and UWB applications. (iii) It generates multiple resonance frequencies, which are also necessary for high-resolution imaging and better penetration in the deeper portion of the object in MWI applications. In MWI applications, microwave excitations with higher resonance frequency components are necessary to achieve a high resolution for reconstructing images but this contradicts the skin depth effects, which restrict the penetration of the higher frequency into biological tissues due to the high conductivity, associated with the water content of those tissues. Therefore, the usage of multiple resonance frequencies is quite significant to take benefit of both high resolution and better penetration to diagnose the deeper objects (i.e., tumor) [8]. Besides, high resonance frequency produces more data set for high-resolution image reconstruction due to better penetration. (iv) the antenna has higher fidelity factor that confirms the lower alteration of the transmitted signal, which is the requirement for MWI applications. Therefore, in the antenna design, to get all features of the proposed antenna, we have used the three octagonal, rectangular slotted ring-shaped parasitic resonator components and different categories slotting technique on the ground plane to enhance the antenna performance, gain, bandwidth and directivity of the radiation pattern. The operating frequency band of the antenna is 2.80 GHz to 11.50 GHz. It is seen that three resonance frequencies are generated at 3.50 GHz, 6.45 GHz, and 10.20 GHz due to applying different slotting techniques that are helped to produce the high-resolution image for MWI applications. The antenna has a maximum of 5.80 dBi gain with a highly directive radiation pattern. Moreover, to validate antenna transmission and reception signal response, three setups (i.e., side-by-side (*X*-axis) scenario, side-by-side (*Y*-axis) scenario, and face-to-face scenario) have been investigated and their corresponding group delay was measured, which is essential for the microwave imaging system. Furthermore, measured fidelity factor (FF) for the face-to-face scenario of the proposed antenna is 0.9091, which is higher than the remaining two scenarios. Measured and simulated outcomes of both time domain and frequency domain demonstrate that the proposed antenna is an appropriate candidate for the microwave imaging applications. The aforementioned features of the antenna are described in detail in the different sections in this paper. The remaining part of the paper is organized as follows. Section 2 describes the design methodology of the proposed antenna. Antenna parametric studies are discussed in Section 3. Antenna performance analysis (simulated and measurement results) in both frequency and time domains is discussed in Section 4. Finally, the conclusion is presented in Section 5.

## 2. Design Methodology of the Proposed Antenna

The ultimate goal of a UWB antenna design is to classify the difference in dielectric constant value among the hidden objects in MWI applications. A UWB antenna must have the capability to generate multiple resonance frequencies in MWI applications because multiple resonance frequencies are essential for high quality-full resolution imaging as well as better penetration in the deeper part of the object (i.e., breast, the human brain, etc.). In addition, the antenna also must have wider bandwidth, high gain, directive radiation pattern as well as high efficiency. However, it is observed that the new parasitic resonator-based UWB antenna can achieve the mentioned properties. Hence, in this paper, a new octagonal ring-shaped parasitic resonator-based compact UWB antenna has been designed and explained its detail features. The geometrical structure of the proposed antenna and fabricated prototype are presented in Figure 1 and Figure 2, respectively. The antenna is designed on the minimal cost epoxy resin fiber (FR-4) dielectric substrate material with a thickness (*h*) of 1.5 mm and having 4.3 relative permittivity ($\varepsilon_{r}$) as well as loss tangent (δ) is 0.02. Generally, the design phase begins with the selection of the lowermost operational frequency for the antenna. Typically, the lower frequency of the antenna operation depends on both antenna width (W) and effective dielectric constant ($\varepsilon_{eff}$) of the material. However, antenna width (W) and length (L) are calculated by using the following formulas [37]:(1)$$W=\frac{c}{2f_{0}\sqrt{\frac{\varepsilon_{r}+1}{2}}}$$
(2)$$\varepsilon_{eff}=\frac{\varepsilon_{r}+1}{2}+\frac{\varepsilon_{r}-1}{2}\;\left[\frac{1}{\sqrt{1+12\left(\frac{h}{W}\right)}}\right]$$
(3)$$L=\frac{c}{2f_{0}\sqrt{\varepsilon_{eff}}}-0.824h\left(\frac{\left(\varepsilon_{eff}+0.3\right)\left(\frac{W}{h}+0.264\right)}{\left(\varepsilon_{eff}-0.258\right)\left(\frac{W}{h}+0.8\right)}\right)$$Here, *c* represents the speed of light in free space, $f_{0}$ represents the resonance frequency. The overall compact dimension of the proposed antenna is 29 × 24 × 1.5 mm^3^. The initial dimension of the proposed parasitic resonator based UWB antenna is taken based on the patch antenna design Equations (1)–(3). After that, we have used 3D electromagnetic simulator CST to optimize the design parameters for achieving the required impedance bandwidth, gain and efficiency. The diamond-shaped patch and two octagonal, rectangular slotted ring-shaped parasitic elements are designed on top of the substrate and then, one octagonal, rectangular slotted ring-shaped parasitic element and partial slotting ground plane are designed on the backside of the substrate. In this antenna design, the diamond-shape with an incremental staircase patch is working as the main radiating element. One octagonal, rectangular slotted ring-shaped parasitic element is placed in the middle position of the backside of the substrate for increasing the directivity of the radiation of the antenna. The width of the ground plane is 24 mm, and the height is 7 mm. The feed line width is 3.00 mm and 9.8 mm in height, and it is directly linked to the main radiating patch as well as was fed with a 50 Ω SMA (subminiature version A) connector. The dielectric constant and electrical conductivity of the SMA connector is 2.08 and 4.62 × 104 S/m, respectively. These components are tweaked in an intelligent way to attain the required antenna performance. In addition, two octagonal, rectangular slotted ring-shaped parasitic elements are situated alongside the feed line to enhance the reflection coefficient, and a 0.5 mm gap has reserved to increase the bandwidth. This gap reduces the effect of the inductive reactance and increases the capacitance at the feed point, which is the main physics to increase the bandwidth. The width of the feed line, the presence of octagonal, rectangular slotted ring-shaped parasitic elements as well as five staircases of the antenna have a strong effect on the matching of the impedance. The parasitic elements are also helped to increase the resistance and reduce the reactance of the antenna in the lower frequencies. As a result, the gain in the main lobe is increased and the side lobe level is concurrently reducing. It is investigated that an octagonal ring-shaped parasitic element properly distributed surface current and help to comparatively reduce the radiation loss instead of other shapes [38]. The staircase diamond-shaped radiating patch is electronically coupled with the ground plane and parasitic elements, and it further enhances the bandwidth. In this work, we have designed a slotted resonator based UWB antenna to achieve the UWB features. The slotting technique is one of the popular techniques to enhance the antenna performance such as gain, radiation directivity, and bandwidth of the antenna [28]. The parasitic resonator also helps to enhance the antenna features such as gain, radiation directivity, and bandwidth of the antenna [9]. In this work, the idea of the slotting technique is inspired by the fractal antenna design [39]. In the proposed antenna design rectangular-shaped and staircase zigzag-shaped slots are used which are located on the radiating patch, parasitic elements, and ground plane as the current distribution observed at the highest concentration. A set of calculations has been carried out for each slot on the radiating patch, parasitic elements and ground plane of the antenna. As a result, the improvement slotted design has created an enhancement in antenna performances such as reflection coefficient, gain, efficiency, and radiation directivity [28]. In general, the traditional antenna design such as patch and ground is more likely characterized as the resistor, while the different types of the slot are defined as an RLC resonator [40]. The different types of slot designs are useful in controlling the antenna’s performances [41]. Furthermore, the slots have been used to diminishing the current flow at any region of the ground or patch. In general, the resonant wavelength of each slot can be calculated by using the following formula:(4)$$\lambda_{0}=\frac{N\times L_{tot}}{\sqrt{\frac{1+\varepsilon_{r}}{2}}}$$
where, $\lambda_{0}$ represents the resonant wavelength of each slot, $\varepsilon_{r}$ represents the relative permittivities, *N* is the total number of slots, and $L_{tot}$ represents the total length of the slots. In the proposed design, three rectangular-shaped slots have been cut out at the top of the ground due to increasing the antenna bandwidth and gain. These shape variations the inductance and capacitance of the input impedance of the antenna, which in turn leads to a change in bandwidth, reflection coefficient, radiation directivity, and properly distributed the current and changes the current movement in the ground. For increasing radiation directivity, efficiency, and gain, two staircase-shaped zigzag slots have been used. In this design, four polygon curves are used for creating two irregular staircase-shaped zigzag slots. On the left and right side of the ground plane, there are two zigzag curves; one is the inner curve, and the other is the outer curve. The two irregular staircase-shaped zigzag slots have been cutout at the bottom of the ground to enhance the radiation directivity, bandwidth, and gain. Moreover, these types of slots have produced an effective length of the diversion of current density to flow in the direction on the ground and allow the antenna prototype radiated directionally [28]. After that, at the left and right corner of the bottom of the ground have been chamfered. For chamfering the current in the corner region of the ground changes their direction as diagonally and due to the fact, the upper operating frequency band is increased.

The proposed antenna parameters of the final design are demonstrated in Table 1. It is seen that in the geometric layout of the antenna, there are 28 parameters, which are used to explain the complete antenna structure. The overall width and length of the antenna are denoted by *W* and *L* and, their values are considered as 24 mm and 29 mm, respectively. Three rectangular-shaped slots each has been cut out at the top of the ground plane of the antenna with the height and width denoted by *h_2_* and *s*, as well as the distance between two slots, is denoted by *d* due to enhance the reflection coefficient under −10 dB and improve the gain of the antenna, because these shapes can adjust the electromagnetic coupling effects between the radiating patch and the ground plane and improve the bandwidth and gain of the antenna [14]. The length of the upper edges of the diamond-shaped patch is denoted by *a*. The value of the parameter *a* is 13.31 mm. A length of 4 mm rectangular-shaped box is designed in the middle position of the patch, which is denoted by parameter *b* and merged with the top diamond-shaped patch. After that, five rectangular-shaped boxes are designed. The length of all boxes is the same (i.e., 1 mm) and denoted by *c*, but different width. These are attached and placed on top of the substrate and then merged with the upper part of the patch. As a result, the radiating patch looks like a staircase diamond-shaped patch. The length and width of the feed line are denoted by *f* and *T*. The values of *f* and *T* are 9.8 mm and 3 mm respectively. It is directly connected with the patch for the excitation of the antenna. A rectangular-shaped slot of 0.5 mm width and 2 mm length has been cut out from both sides of the feed line and all corner edges have been blended. Two same-sized octagonal, ring-shaped parasitic elements are placed alongside the feed line. Both octagonal parasitic elements have three different arm lengths, and these lengths are denoted by *l_1_, l_2_*, and *l_3_*, respectively. The considered values of *l_1_, l_2_*, and *l_3_* are 3.62 mm, 2.62 mm, and 3.11 mm, respectively. A rectangular-shaped slot length and width are denoted by *h_1_*, and *g* and their values are 1.5 mm and 2 mm, respectively. The slot has been cut out at the top-middle position from both parasitic elements. On the other hand, one octagonal, ring-shaped parasitic element, and partial slotted ground have been designed on the backside of the substrate. One large-sized octagonal, ring-shaped parasitic element is positioned in the middle of the substrate. Its lengths of the arm are denoted by *l_4_, l_5_*, and *l_6_*, as well as values, which are 5.42 mm, 4.67 mm, and 2.42, respectively. A rectangular-shaped slot length and width are denoted by *h_3_* and *g*, and their values are 1 mm and 2 mm respectively. This rectangular-shaped slot has been cut out at the top-middle position of the large-sized octagonal parasitic element. A slotted partial ground plane has been designed on the backside of the substrate. The width is denoted by *W* (24 mm) and length is denoted by *H* (7 mm) of the ground plane. For enhance the reflection coefficient under −10 dB and improve the gain of the antenna three same sized rectangular-shaped slots have been cut out at the top of the ground plane. The distance (*d*) between two slots is 6.72 mm. The length and width of the rectangular-shaped slot are denoted by *h_2_* and *s*, and values are 2.25 mm and 2 mm, respectively. Then all corner edges of the rectangular-shaped slot have been blended. After that, a length of 2.83 mm, which is denoted by *x*, has been chamfered at the left and right corner of the bottom of the ground. Moreover, two irregular zigzag slots have been cut out on the ground plane. The four polygon curves are used for creating two irregular zigzag slots. On the right side of the ground plane, there are two zigzag curves, one is the inner curve, and the other is the outer curve. The width of the outer curve is denoted by *w_1_*, and its value is 6.20 mm, and the width of the inner curve is denoted by *w_2_*, and its value is 5.35 mm. Therefore, the final width of the right-side zigzag slot is 0.85 mm. In contrast, on the left side of the ground plane, there are also two zigzag curves, one is the inner curve, and the other is the outer curve. The width of the outer curve is denoted by *w_3_*, and its value is 5.55 mm, and the width of the inner curve is denoted by *w_4_*, and its value is 4.80 mm. So, the final width of the left side zigzag slot is 0.75 mm. But due to irregular phenomena of the zigzag slots, the widths are not the same in both cases. Zigzag slots are made of four staircases with different height and width. The bottom staircase width is denoted by *p* and value is 0.80 mm in both cases, but the top staircase widths in both zigzag slots are denoted by *i* and *t*, and their values are considered as 0.80 and 0.90 mm respectively.

## 3. Parametric Study of the Proposed Antenna

The proposed resonator-based UWB antenna constructs a diamond-shaped radiating patch and a ground plane fronting 180° in respect of each other. For a simple explanation, including the final design with several modification sequences of the antenna are illustrated in Figure 3. The width and length of the prototype are denoted by *W* and *L*, respectively. A slotted rectangular-shaped plane linked to the ground. The length and width of this shape are denoted by *W* and *H*. The height of the staircase slots of the patch is denoted by *c*. Three rectangular-shaped slots each has been cut out at the top of the ground plane of the antenna with the height and width denoted by *h_2_* and *s*, as well as the distance between two slots, are denoted by *d* due to enhance the reflection coefficient under −10 dB and improve the gain of the antenna. The staircase-shaped zigzag slots have been used to enhance the gain and radiation directivity of the antenna [28]. In this work, four staircase-shaped two irregular zigzag slots have been cut out at the lower portion of the ground plane. It is called irregular because the size, length, staircase shape widths are different. The bottom staircase width of both zigzag slots is the same, and it is denoted by *p*, but top staircase widths are different, which are denoted by *i* and *t*. The main goals of these irregular staircase-shaped zigzag slots are (i) to enhance the reflection coefficient under −10 dB, (ii) to produce three resonance frequencies under −20 dB (iii) to increase the antenna gain and (iv) to improve the bandwidth for UWB antenna. With the help of four steps staircase-shaped zigzag slot edges on the ground plane, improve the radiation directivity due to the proper movement of the surface current distribution. It is investigated that if only one irregular zigzag slot is used, then the reflection coefficient curve is changed, and it is far away from the −10 dB, as well as gain, is decreased, because of the lack of proper movement of surface current. The two octagonal, rectangular slotted ring-shaped parasitic elements have been placed alongside the feed line. The lengths of arms of both parasitic elements are denoted by *l_1_, l_2_*, and *l_3_*. The height and width of the middle rectangular-shaped slot of both parasitic elements are denoted by *h_1_* and *g*, respectively. Furthermore, one octagonal, rectangular slotted ring-shaped parasitic element has attached in the middle of the backside of the substrate, and length of arms are denoted by *l_4_, l_5_*, and *l_6_*. The height of the middle rectangular-shaped slot of the backside parasitic element is denoted by *h_1,_* and the width is the same as *g*.

The parasitic resonator also helps to improve the antenna features such as gain, radiation directivity, and bandwidth of the antenna [9]. Chamfering is one kind of technique that is used to enhance the operating frequency band [42]. In this study, it is observed that at the left and right corner of the bottom of the ground plane have been chamfered, which length is denoted by *x* for increasing the frequency band from 11.00 GHz to 11.50 GHz. It is happening due to changing the current movement path on the ground [42]. However, these cuts have a momentous effect on the surface current distribution, which assists in attaining an overall frequency band from 2.80 GHz to 11.5 GHz.

Figure 4 illustrates the effect of the octagonal, rectangular slotted ring-shaped parasitic elements and ground slotting on the reflection coefficient (i.e., S_11_ parameters) and maximum gains according to the design sequence of Figure 3. At first, all slots etched from the patch, parasitic elements, and ground plane as mentioned size in Table 1. After that, we analyzed and investigated the antenna parameters. Lastly, a satisfactory outcome is attained with the modifications which show the desired features of the UWB antenna. From Figure 4a, the overall observation demonstrates that the proposed prototype has extensive bandwidth compared to the other verified shapes, including normal patch slotted design, with slotted front resonator design, with slotted back resonator design and partial ground slotted designs. Firstly, in normal patch slotted design (i.e., Figure 3a), the operating frequency is 2.78 GHz to 7.34 GHz with one resonance frequency under −20 dB, but the reflection coefficient curve is very close to −10 dB at 4.64 GHz. Secondly, when two octagonal, rectangular slotted ring-shaped parasitic resonator elements are placed alongside the feed line of the radiating patch (i.e., Figure 3b), then generate two resonance frequencies under −20 dB. The beginning frequency has shifted from 2.78 GHz to 2.79 GHz, and higher frequency has shifted to 7.31 GHz, so bandwidth is slightly decreased, but it helps to increase the radiation directivity. In this case, the overall bandwidth is 2.79 GHz to 7.31 GHz. Thirdly, when two octagonal, rectangular slotted ring-shaped parasitic resonator elements are placed alongside the feed line of the radiating patch, and one octagonal, rectangular slotted ring-shaped parasitic resonator element placed in the middle of the substrate (i.e., Figure 3c), then generates one resonance frequency providing a reflection coefficient lower than −23 dB and generates another resonance frequency providing a reflection coefficient lower than −28 dB. In this way, the new antenna’s operating frequency is 2.79 GHz to 7.32 GHz. So, the upper frequency is slightly increased as well as it also helps to increase the radiation directivity, but it does not cover the UWB. In this scenario, it is observed that the reflection coefficient curve is far away from the −10 dB between 7.32 GHz to 11.18 GHz; however, this problem is partially solved by design four (i.e., Figure 3d). By applying three rectangular slots on top of the ground plane, the beginning frequency has shifted to a lower frequency of 2.79 GHz and then generated two resonance frequencies; the first one resonance frequency provides a reflection coefficient lower than −20 dB and another under −26 dB. It means that this modification also helps to enhance the radiation directivity, as well as the reflection coefficient, which is slightly lower −10 dB. However, in this scenario, the overall bandwidth is 2.79 to 7.33 GHz, but it also does not cover UWB. Also, in the proposed design (i.e., final design (Figure 3e)), by applying two irregular staircase shaped zigzag slots and chamfered the left and right corner of the ground plane, then generated three resonance frequencies providing reflection coefficients lower than −10 dB; the first one is under −20 dB, the second one is under −33 dB and the third one is under −31 dB. Therefore, the lower frequency begun from 2.80 GHz as well as the upper frequency is shifted to 11.50 GHz. Consequently, it extended the operational upper frequency by about 4.18 GHz (7.32 to 11.50 GHz), as well as produced overall operating frequency from 2.80 GHz to 11.50 GHz with a reflection coefficient lower than −10 dB, which is covered entire UWB. However, the use of parasitic elements and different types of slots generates some additional current-conducting paths. This modification, variation the inductance and capacitance of the input impedance of the antenna, which indicates changes in the prototype features. The simulated maximum gain curves for the several modifications are illustrated in Figure 4b. At a glance, the maximum gains for the different modification structures are demonstrated in Table 2. The maximum gain of the prototype is 5.8 dBi, whereas the normal patch slotted, with slotted front resonators, with slotted back resonator and partial ground slotted antenna, has a gain of 3.47, 3.89, 4.36 and 4.40 dBi respectively. It is observed that the gain is gradually increased with respect to frequencies from 2.80 GHz to 7.50 GHz, 8.80 GHz to 9.45 GHz, as well as 10.5 GHz to 11.50 GHz. At these frequencies, the gain is increased due to the higher efficiency of the antenna. Furthermore, the gain is slightly decreased between 7.50 GHz to 8.80 GHz as well as 9.45 GHz to 10.50 GHz. The main reason for this incident is that the efficiency is slightly lower, as well as generates some back lobes [9]. Besides, these incidents happened due to applying different types of slots in both patch and ground. The slots distribute the current conduction path and raise the electrical length as well as generating strong directional radiation owing to the mitigation of the surface current [9,14,28]. Therefore, the different modifications have an important effect on the gain and efficiency of the antenna. However, the overall gain is increased, and the maximum gain reached 5.8 dBi at 9.45 GHz. Finally, the performance evaluations of the effects of various antenna design structures/modifications are presented in Table 2.

However, it is shown that if the length of any design parameter of the antenna is changed then the effect has happened on the antenna performance such as reflection coefficient (S_11_), gain, efficiency, etc. For instance, here we investigated only two parameters for a simple explanation. Figure 5 illustrated the effect of the variation of the antenna parameters on the reflection coefficient. When the value of the length of the parameter *a* is 0.81mm decreased (i.e., 12.50 mm instead of 13.31 mm), then the reflection coefficient curve goes to upward in between at 8.50 GHz to 9.50 GHz from −10 dB. Moreover, the lower frequency has shifted from 2.80 GHz to 3.00 GHz and the higher frequency has shifted to 11.50 GHz to 11.00 GHz. In this case, the operating frequency band is 3.00 to 8.61 GHz with three frequencies which is not cover UWB. In contrast, when the value of the length of the parameter *a* is increased 0.81 mm (i.e., 14.12mm instead of 13.31 mm) then the reflection coefficient curve goes to upward in between at 8.40 GHz to 9.45 GHz from −10 dB. Moreover, the lower frequency has shifted from 2.80 GHz to 2.90 GHz and the higher frequency has shifted to 11.50 GHz to 11.54 GHz (i.e., 40 MHz frequency band increased). In this case, the operating frequency band is 2.90 to 8.45 GHz with three frequencies which is not cover UWB. Therefore, it is concluded that the value of the edge’s parameter *a* is optimized when its value is considered as 13.31 mm, and it is covered entire UWB. However, due to the variation of the parameter *a*, the effect on the reflection coefficient is presented in Figure 5a. On the other hand, if the length of the ground plane is 1 mm increased (i.e., H = 8 mm) then, the reflection coefficient curve goes to upward in between at 7.50 GHz to 8.48 GHz and 10.80 GHz 11.00 GHz from −10 dB. In addition, the lower frequency has also shifted from 2.80 GHz to 3.00 GHz and its operating frequency band is 3.00 GHz to 7.69 GHz with two resonance frequencies under −10 dB that is not also the UWB. If the length of the ground plane is 1 mm decreased (i.e., H = 6 mm) then, the reflection coefficient curve is very close to −10 dB in between 4.28 GHz and 4.64 GHz, as well as this curve also goes to upward in between at 7.00 GHz to 7.91 GHz and 8.44 GHz 9.68 GHz from −10 dB. In addition, the lower frequency has also shifted from 2.80 GHz to 3.00 GHz and its operating frequency band is 2.81 GHz to 7.00 GHz with two resonance frequencies providing reflection coefficients lower than −10 dB which is not cover UWB. So, the value of the ground plane parameter *H* is optimized when its value is considered as 7 mm, and hence it is covered entire UWB. Therefore, due to the variation of the parameter *H*, the effect on the reflection coefficient is presented in Figure 5b. Finally, it is also indicated that the considered values a = 13.31 mm and H = 7 mm are the optimized value for the proposed prototype. Therefore, the different modifications have an important effect on the gain and efficiency of the antenna.

Figure 6 is illustrated the statistical surface current delivery of the prototype for three resonance frequencies such as (i) 3.50 GHz, (ii) 6.45 GHz, and (iii) 10.20 GHz. In this study, we have used 3D CST microwave studio 2018 simulator software for observing the surface current. From Figure 6, it is seen that the maximum prevalent surface current conducting region of the antenna is around the feeding line as well as the bottom part of the radiating patch, but at 6.45 GHz frequency, the moderate current conduction region is also the upper part of the patch as well as around the different slotting on the ground. Furthermore, at 10.20 GHz frequency, there exist a small number of nulls on the radiating patch due to the upper order current mode and adequate current movements around the different slots on the ground plane. The existence of the rectangular slots and irregular zigzag slots as well as the presence of the octagonal ring-shaped parasitic element alters the current movement path and modifies the antenna features, particularly to expand the upper range of the operating frequency. However, for attaining the wide-ranging frequency band the antenna sustains the harmonic order flow in the radiating patch and the ground plane.

## 4. Antenna Performance Analysis

The proposed prototype’s performance has been analyzed and simulated by using CST microwave studio 2018 simulator software. Moreover, plotting the simulated results, the data analysis software Origin Pro 2018 (OriginLab Corporation, MA, USA) has been used. Furthermore, by using the PNA network analyzer (Agilent Technologies, Inc., N5227A 10 MHz–67 GHz) is used to measure the reflection coefficient (S_11_) parameters, and the setup is illustrated in Figure 7a. On the other hand, the UKM Satimo near field StarLab (Microwave Vision Group, Paris, French) is used to measure the radiation pattern, efficiency, as well as gain of the fabricated prototype. The mechanism is used for measuring the electric fields (EF) of the prototype in the near field zone because of calculating the corresponding far-field quantities of the AUT (Antenna under test). The AUT is situated in the central point of a circular “arch” and positioned on the testbed that comprises sixteen distinct receiving antennas. In this system, the antennas are positioned in a ring-shaped model with maintaining an equal distance. Generating a full three dimensional (3D) scan, the AUT is rotated 360 degrees horizontally, and after that, we get the three dimensional (3D) radiation pattern. Figure 7a illustrates the PNA measurement set up, and Figure 7b illustrates the Satimo StarLab setup. Through the SatEnv software, the near field data are transformed into far-field data. Furthermore, from the far-field data, the antenna gain, efficiency, and radiation pattern are calculated by using the SatEnv software.

### 4.1. Frequency Domain Characteristics

In this domain we discuss different characteristics of the realized antenna. Figure 8 demonstrates the simulated and measured reflection coefficient of the realized prototype antenna. The proposed antenna achieves overall −10 dB impedance bandwidth of 8.7 GHz (2.80 GHz–11.50 GHz). The overall bandwidth formed the antenna appropriate for the MWI applications. From the simulated result, it is shown that there existed three resonance frequencies wherein, the first one shows a level of −20 dB on the reflection coefficient that was a lower resonance, the second one is a level of −35 dB on the reflection coefficient that was a higher resonance, as well as the third one is a level of −31 dB on the reflection coefficient that was also a higher resonance, but slightly lower than the second one. On the other hand, it is observed from the measured result, and the first resonance frequency has generated at 3.49 GHz, second resonance has generated at 6.85 GHz, which is slightly shifted (i.e., 400 MHz) toward upper frequency as well as third resonance frequency has generated at 10.24 GHz, which is also little bit shifted (i.e., 40 MHz) toward upper frequency. Therefore, it is concluded that the measured and simulated results made a good agreement. Besides, low reflection coefficient the advantage of having resonance frequencies for the microwave imaging applications is that the transmission coefficient is increased. As a result, the microwave signal penetration depth is increased and it is passed easily through the tissues (i.e., breast, head, skin, etc.) in microwave imaging applications. Another advantageous of low reflection coefficient for microwave imaging application is low noise generation during the reflection from high dielectric tissue like tumour object. The simulated and measured antenna’s gain versus the frequency is demonstrated in Figure 9a, and radiation efficiency is shown in Figure 9b. It is seen, the result shows that the proposed prototype retains a more than 5.7 dBi gain over the whole frequency band, and its maximum gain is 5.8 dBi at 9.45 GHz. It is seen that the measured and simulated outcomes are consistent. The proposed prototype achieves a better gain being compact configuration related to other parasitic resonator-based antennas. Remarkably, that the higher gain antenna is more suitable for UWB applications as well as MWI applications. Therefore, the proposed prototype is an appropriate candidate for the use of MWI applications due to its higher gain. Figure 9b illustrates the simulated and measured radiation efficiency of the prototype against frequency. From Figure 9b, in both simulated and measured results, we can observe that the proposed antenna’s average radiation efficiency is approximately 75%, with a maximum of 82% over the entire bandwidth because of the higher gain of the antenna. Therefore, it is decided that the measured and simulated results made a good agreement. It is witnessed that the use of improved structure together with ring-shaped parasitic resonator elements on the substrate, different types of slotting on the ground plane and stairway diamond-shaped radiating patch generates few additional excitations, and consequently, the operating frequency band is increased.

Figure 10 demonstrated two dimensional (2D) and three dimensional (3D) radiation patterns, including the measured and simulated cross-polarization and co-polarization of three resonance frequencies of 3.50 GHz, 6.45 GHz, and 10.20 GHz respectively. The Theta (θ) and Phi (φ) spherical coordinates are associated with the Cartesian axis alignments such as, when Phi = 0° then, Theta (θ) = 0° to 360° degrees is called the XZ cut, and when Phi (φ) = 90°, Theta (θ) = 0° to 360° degrees is known as YZ cut, as well as when Theta (θ) = 90°, Phi (φ) = 0° to 360° degrees is assumed the XY cut. The YZ-plane (φ = 90°) is called as H-plane as well as the XZ-plane (φ = 0°) is called as E-plane. From the near field measurement, it is remarkable that the proposed prototype is directional, and the main radiation direction is towards. The significant lobes of the radiation patterns are permanent towards the end-fire direction over the whole frequency band. The proposed prototype demonstrates a steady directional radiation pattern over the entire operating band and resonance frequencies, but at higher frequencies, the radiation of the antenna is slightly omnidirectional with some back lobes. However, at the frequencies 3.50 GHz, 6.45 GHz, and 10.20 GHz, the antenna exhibitions a distinctive eight shapes of the radiation pattern. Moreover, at a higher frequency, the prototype demonstrates directive cross-polarization due to fluctuating the current distribution as well as some slots. Besides, at higher frequency (i.e., 10.20 GHz) frequency, the current does not dispense consistently owing to the upper sequence current mode excitation that produced the radiation directive. Furthermore, at upper frequencies, several nulls may rise in current distribution that generates few back lobes.

### 4.2. Time Domain Analysis

Figure 11 illustrated three types of setups for the measurement of the time-domain performance of the antenna. The first one is face to face (F2F); the second one is side-by-side (SbS) (*X*-axis), and the third one is side-by-side (SbS) (*Y*-axis). For measurement purposes, the distance between the antennas is considered as 240 mm for three scenarios. It is observed that, overall, in three scenarios, the waveforms are almost similar for transmitted and received pulses, except it spread marginally. As a result, it is concluded that the antenna can radiate a short pulse with minimal alteration. However, for SbS (*Y*-axis) (i.e., Figure 11b) and F2F (i.e., Figure 11c) scenarios, the received waveforms are similar to the transmitted pulses, due to the high directivity of the antenna. On the other hand, for SbS (*X*-axis) (i.e., Figure 11a) scenario, the received and transmitted pulses are not the identical precisely, because of the antenna’s directional radiation characteristics. Aimed at directional radiation, the side radiation a little bit distorted as well as the inequality is observed among the received and transmitted signals. Therefore, received and transmitted magnitudes are not of equivalent shape. It is known that the term fidelity factor is considered to validate the correlation between the transmitted and received pulses. It means that the highest magnitude of the cross-correlation between the transmitted and received signals. The fidelity factor (FF) is calculated in the MATLAB programming platform by using the following formula:(5)$$FF=Max\frac{\int_{-\infty }^{+\infty }m\left(t\right)n\left(t-\tau \right)dt}{\sqrt{\int_{-\infty }^{+\infty }\left|m{\left(t\right)}^{2}\right|dt\;\int_{-\infty }^{+\infty }\left|n{\left(t\right)}^{2}\right|dt}}$$

Here, m(t) denotes the transmitted signals; n(t) denotes the received signals. The fidelity factors for SbS (*X*-axis) scenario, SbS (*Y*-axis) scenario, and F2F scenario are 0.8283, 0.8270 and 0.9091 respectively. However, the higher value of the fidelity factor confirms a small alteration of the transmitted signal, which is the compatible for MWI applications. In the time domain performance, the term “group delay” is a prominent term that represents the signal phase distortion. The simulated and measured group delay is illustrated in Figure 11d. From Figure 11d, it is observed that the group delay of F2F is approximately steady except a little bit distortion between 5.20 GHz to 5.90 GHz and 8.80 GHz to 9.20 GHz. It may be happened due to some noise in measurement setup, otherwise, it is nearly continual over the entire frequency. Therefore, it is concluded that this kind of arrangement is recommended for the MWI system. However, the comparison of specified antennas with the proposed approach is described in Table 3. In this case, the assumed measurable factors are dimension, operating frequency, bandwidth, gain, and applications. Finally, it may be remarked that the proposed prototype has a smaller size, wider bandwidth, better gain, and better efficiency than the specified antennas.

## 5. Conclusions

In this paper, a complete and compact octagonal ring-shaped parasitic resonator-based antenna is designed and developed, having a dimension 29 × 24 × 1.5 mm^3^. The performance is investigated and concluded that this antenna is suitable for microwave imaging systems. The antenna is verified in both the time domain and the frequency domain with proper characterization. The proposed prototype attains satisfactory fractional bandwidth (FB) over the Ultrawideband of 121.67% (2.80 GHz to 11.50 GHz) with higher gain, a stable directional radiation pattern, and almost larger efficiency. This prototype has achieved a maximum of 5.80 dBi gain in the operating band. In the antenna design, owing to improving the overall bandwidth, radiation pattern, gain, and efficiency, the various antenna measurable factors and parameters are optimized as well as used octagonal, rectangular slotted ring-shaped resonators comprising different kinds of slots on both radiating patch and the ground plane. The antenna also illustrations superb time-domain performance on the F2F and SBS (*Y*-axis) setups. It is observed that for F2F setup, the proposed antenna prototype has a higher fidelity factor (FF) and lower group delay, which are significant factors for MWI. At last, the observation throughout the investigation of the simulation and measured data discloses that the proposed prototype is appropriate for Ultrawideband based MWI applications.

## Figures and Tables

**Figure 1 sensors-20-01354-f001:**
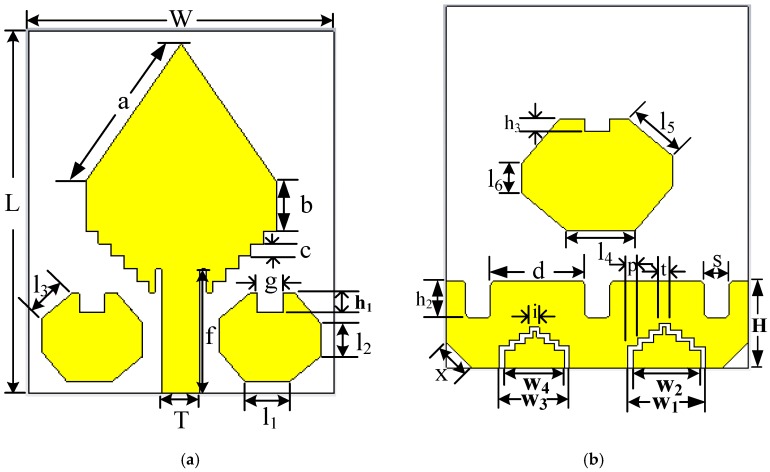
Geometric layout of antenna: (**a**) Topside view; (**b**) Backside view.

**Figure 2 sensors-20-01354-f002:**
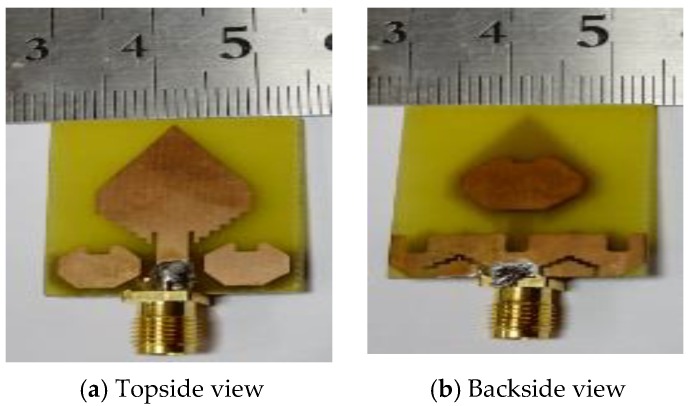
Fabricated prototype: (**a**) Topside view; (**b**) Backside view.

**Figure 3 sensors-20-01354-f003:**
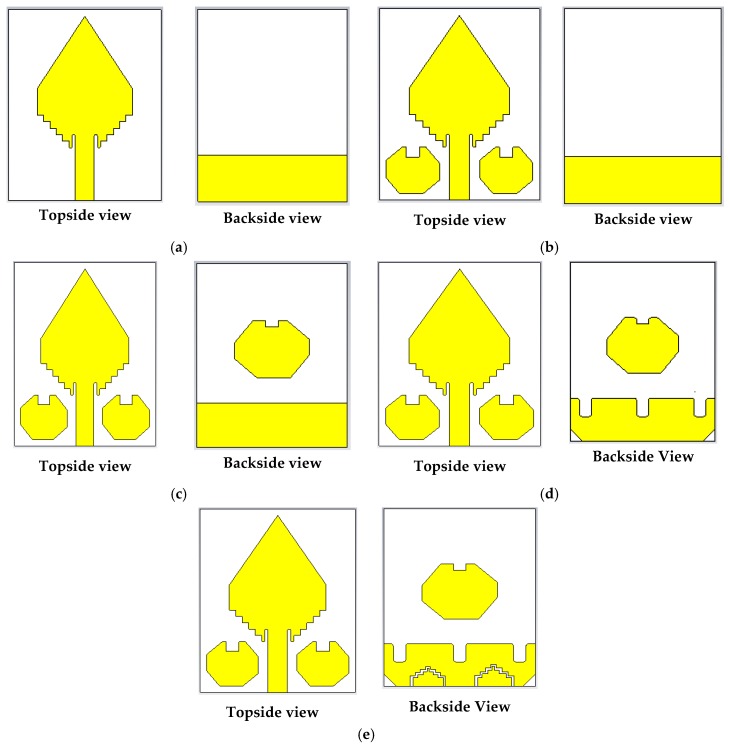
Antenna layout design: (**a**) Normal patch antenna; (**b**) Antenna with slotted front resonators; (**c**) Antenna with slotted front and back resonators; (**d**) Antenna with slotted resonators and partial ground slotting; (**e**) Proposed antenna.

**Figure 4 sensors-20-01354-f004:**
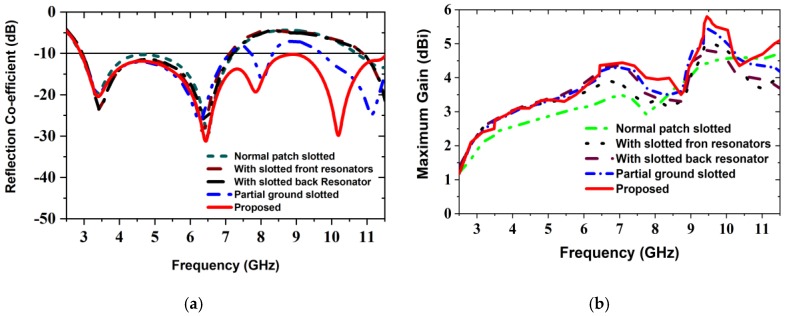
Effect of the various modifications of the antenna: (**a**) Simulated reflection coefficient (S_11_); (**b**) Simulated maximum gain.

**Figure 5 sensors-20-01354-f005:**
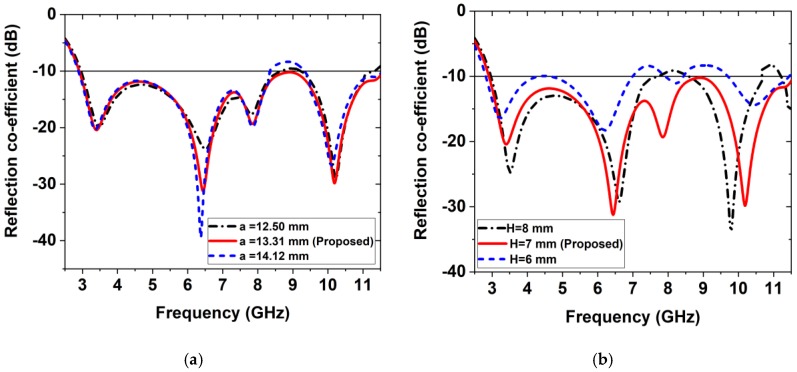
Effect of the variation of the antenna parameters on reflection coefficient (S_11_): (a) When a = 12.50 mm, 13.31 mm, and 13.73 mm; (b) When H = 8 mm, 7 mm, and 6 mm.

**Figure 6 sensors-20-01354-f006:**
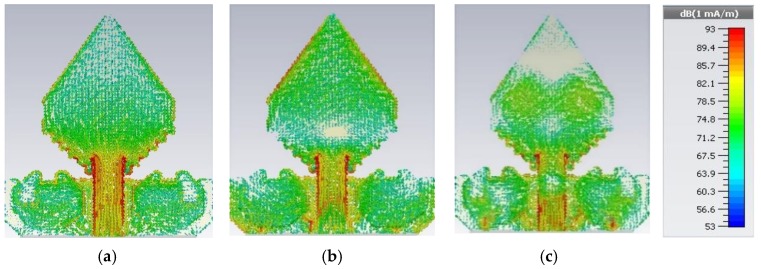
Simulated surface current distribution at: (**a**) 3.50 GHz; (**b**) 6.45 GHz; and(**c**) 10.20 GHz.

**Figure 7 sensors-20-01354-f007:**
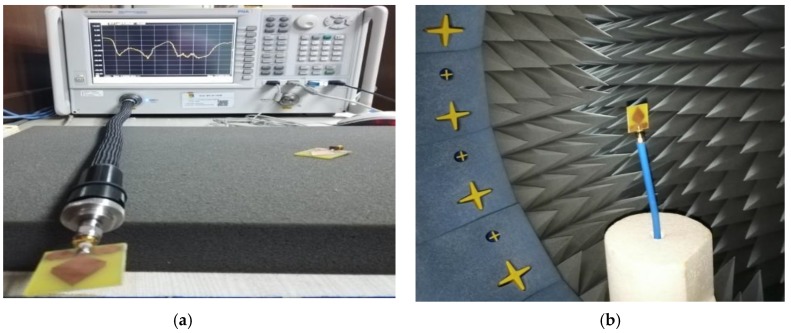
Measurement setup: (**a**) PNA network analyzer measurement setup; (**b**) Satimo StarLab setup.

**Figure 8 sensors-20-01354-f008:**
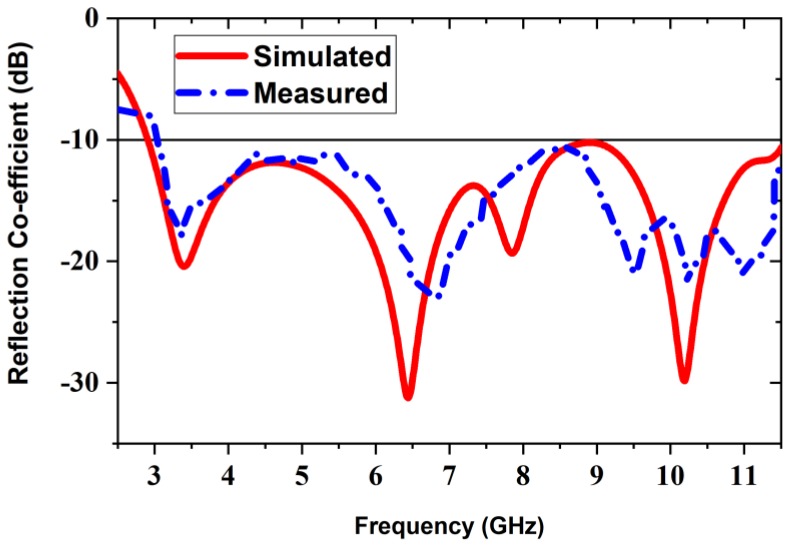
Measurement and simulated reflection coefficient (S_11_) of the proposed antenna.

**Figure 9 sensors-20-01354-f009:**
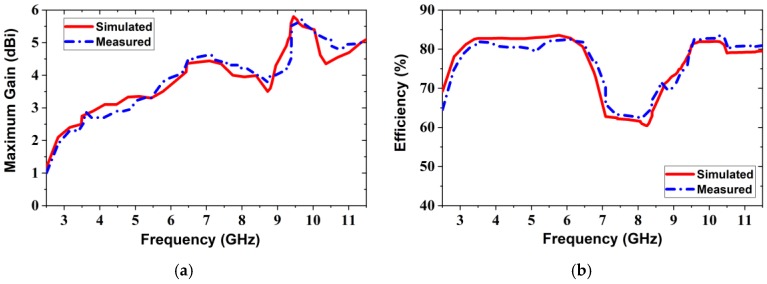
Simulated and measured: (**a**) Maximum gain; (**b**) Radiation efficiency over the frequency.

**Figure 10 sensors-20-01354-f010:**
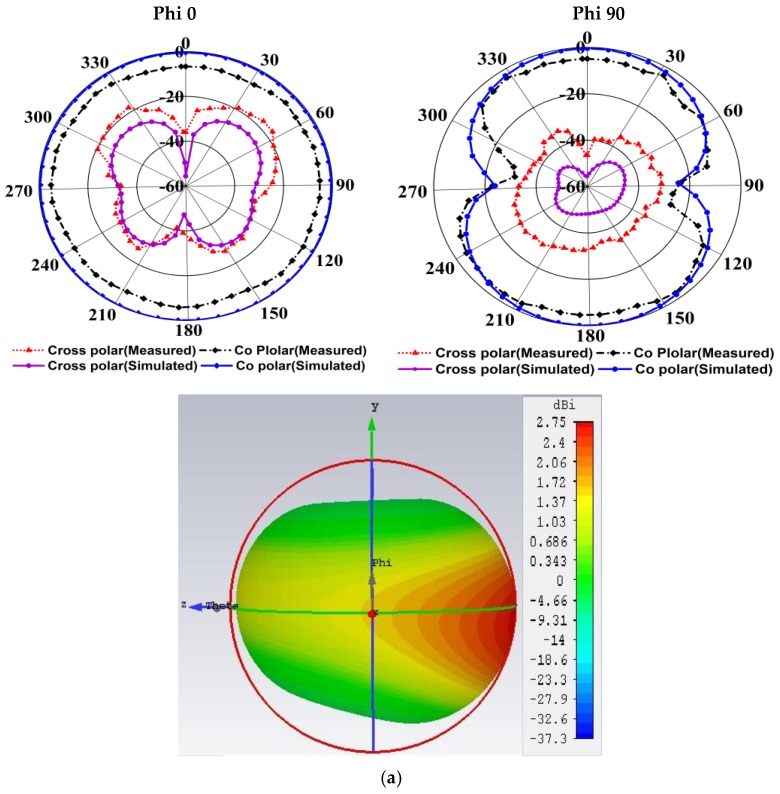
Two dimensional and three-dimensional radiation patterns of the prototype at different frequencies: (**a**) 3.50 GHz; (**b**) 6.45 GHz; (**c**) 10.20 GHz.

**Figure 11 sensors-20-01354-f011:**
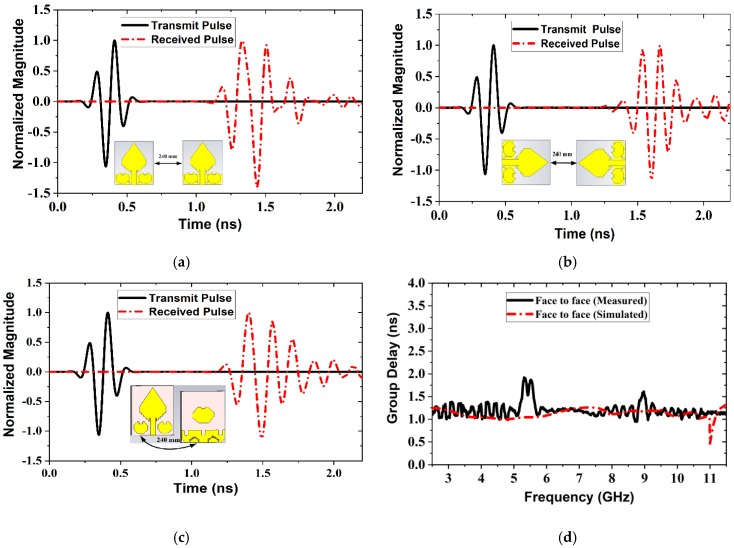
Normalized magnitude of three different setups: (**a**) Side by side (SbS) *X*-axis; (**b**) Side by side (SbS) *Y*-axis; (**c**) Face to face (F2F); (**d**) Face to face group delay.

**Table 1 sensors-20-01354-t001:** Proposed antenna parameters.

Parameters	Size (mm)	Parameters	Size (mm)	Parameters	Size (mm)
W	24	l_6_	2.42	w_1_	6.20
L	29	f	9.8	w_2_	5.35
a	13.31	g	2	w_3_	5.55
b	4	T	3	w_4_	4.80
c	1	d	6.72	t	0.90
l_1_	3.62	h_1_	1.5	i	0.80
l_2_	2.62	h_2_	2.25	p	0.80
l_3_	3.11	h_3_	1	H	7.00
l_4_	5.42	s	2	-	-
l_5_	4.67	x	2.83	-	-

**Table 2 sensors-20-01354-t002:** The Comparison table of the results on various design structures.

Various Design Structures	Operating Frequency Range (GHz)	Bandwidth (GHz)	Maximum Gain (dBi)
Normal patch slotted	2.78–7.34	4.56	3.47
With slotted front resonators	2.79–7.31	4.52	3.89
With slotted back resonator	2.79–7.32	4.53	4.36
Partial ground slotted	2.79–7.33	4.54	4.40
Proposed	2.80–11.50	8.70	5.80

**Table 3 sensors-20-01354-t003:** Comparison of the overall performance with the specified antennas.

Ref. No.	Antenna Type	Dimension (mm^3^)	Operating Frequency (GHz)	Bandwidth (GHz)	Gain (dBi)	Applications
[9]	Diamond-shaped microstrip patch	30 × 25 × 1.6	2.70–10.30	7.60	5.50	Microwave breast imaging
[12]	Circular monopole patch	32 × 24 × 1.6	3.10–9.20	6.10	Not reported	UWB applications
[11]	Y-shaped planar monopole	36 × 38 × 1.6	2.86–13.30	13.30	10.40	UWB applications
[13]	Open loop microstrip patch	24 × 28 × 1.6	3.00–14.00	11.00	4.20	UWB applications
[14]	Small square-shaped monopole	24 × 22 × 1.6	3.04–11.43	8.39	5.10	Microwave imaging
[26]	CPW-fed microstrip patch	76 × 44 × 1.6	2.90–7.80	4.90	10.00	Microwave imaging
[23]	Antipodal Vivaldi	40 × 40 × 1.6	2.50–11.00	8.50	7.20	Microwave breast imaging
[27]	Balanced slotted antipodal Vivaldi	40 × 40 × 1.6	3.01–11.00	7.99	7.06	Microwave imaging
[16]	Wide slotted UWB patch	21.44 × 23.53 × 1.6	3.49–12.00	8.51	5.70	Microwave Imaging
[28]	Slotted antipodal Vivaldi antenna	42.80 × 57.30 × 1.6	3.60–10.00	6.40	7.60	Microwave imaging
**Proposed**	Octagonal Ring-shaped Parasitic Resonator patch antenna	29 × 24 × 1.5	2.80–11.50	8.70	5.80	Microwave imaging

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
