# Peer review of "An Octagonal Ring-shaped Parasitic Resonator Based Compact Ultrawideband Antenna for Microwave Imaging Applications"

_sensors, 2020, doi:10.3390/s20051354_

Round 1
Reviewer 1 Report
The paper proposes a new design of ultrawideband antenna based on the modification of a kind of patch antenna adding resonators and slot-like structures on the ground plane of the antenna.
The authors provide a good analysis on each step of the design of the antenna.
There is valuable technical information in this work, but it is very difficult to read it. Extensive editing of English language and style is required.
Besides, some technical terms are not correctly used. Below there are some examples:
Throughout the paper the authors are mixing radiation pattern with the curve of gain, and radiation curve (and radiation characteristics in line 151, pg. 5) with the curve of reflection coefficient. In lines 262 and 263 (page 8), the authors are mixing "impedance" with "reflection coefficient". For example, the authors mention "-20 dB impedance". From the context it seems that it should read "... a level of -20 dB on the reflection coefficient..." Many phrases are inintelligible, as in line 345, pg. 13 "The signal phase falsification is ..."
The reviewer would gladly recommend the publication of this paper if the authors edit their work concerning the language (grammar and style).
Reviewer 2 Report
The proposed antenna structure is interesting even of not so innovative, it seems another UWB antenna, the innovations introduced in the antenna structure are not so evident and they should be better detailed in he introduction section.
The design methodology it is not so clear, the antenna seems to be designed by following a trial error procedure.
the author’s should provide more details concerning the obtained antenna parameters
Round 2
Reviewer 1 Report
Thank you for reviewing the paper. It is much more readable. Although there are few minor modifications that may be implemented. Please see the attached pdf document.
Concerning the scientific content, the authors put forward the need of resonance frequencies. Besides providing low reflection coefficient (always good for any antenna application), can you provide another advantage of having resonance frequencies for the applications you mentioned?

Reviewer 2 Report
The work has been improved and the author’s tried to follow the reviewers indications, I have no further suggestions.
